# Safety of Biological Therapies in Elderly Inflammatory Bowel Diseases: A Systematic Review and Meta-Analysis

**DOI:** 10.3390/jcm11154422

**Published:** 2022-07-29

**Authors:** Gustavo Drügg Hahn, Petra Anna Golovics, Panu Wetwittayakhlang, Dirlene Melo Santa Maria, Usiara Britto, Gary Edward Wild, Waqqas Afif, Alain Bitton, Talat Bessissow, Peter Laszlo Lakatos

**Affiliations:** 1Division of Gastroenterology, McGill University Health Centre, Montreal, QC H3G 1A4, Canada; gustavodhahn@gmail.com (G.D.H.); golovics.petra@gmail.com (P.A.G.); wet.panu@gmail.com (P.W.); gary.wild@mcgill.ca (G.E.W.); waqqas.afif@mcgill.ca (W.A.); alain.bitton@mcgill.ca (A.B.); talat.bessissow@gmail.com (T.B.); 2School of Medicine, Graduate Course Sciences in Gastroenterology and Hepatology, Universidade Federal do Rio Grande do Sul, Porto Alegre 90035-002, Brazil; estatistica.consultoria@gmail.com (D.M.S.M.); usiara@yahoo.com.br (U.B.); 3Department of Gastroenterology, Hungarian Defense Forces, Medical Centre, 1062 Budapest, Hungary; 4Unit of Gastroenterology and Hepatology, Division of Internal Medicine, Faculty of Medicine, Prince of Songkla University, Songkhla 90110, Thailand; 51st Department of Medicine, Semmelweis University, 1085 Budapest, Hungary

**Keywords:** inflammatory bowel disease, elderly, biologics, safety

## Abstract

**Background and Aim:** Newer biologics appeared safer in landmark clinical trials, but their safety is understudied in vulnerable populations. The aim of the present study was to perform a systematic review and meta-analysis to assess the safety of available biologicals in the elderly IBD population. **Methods:** We systematically searched PubMed/Medline and conference proceedings between 1 April 1969 and 1 June 2021 to identify eligible studies that examined the safety of biologics in elderly patients with IBD. Of the 2885 articles and 12 congress abstracts identified, 12 peer reviewed papers and 3 abstracts were included after independent evaluation by two reviewers. The identified studies collected safety data on anti-TNF, vedolizumab (VDZ) and ustekinumab (UST). **Results:** Rates of AE and infections were not different among the biologics (AE mean rate: 11.3 (CI 95% 9.9–12.7)/100 pts-years; *p* = 0.11, infection mean rate: 9.5 (CI 95% 8.4–10.6)/100 pts-years; *p* = 0.56) in elderly IBD patients on anti-TNF, VDZ or UST. Infusion/injection reaction rates were more common on anti-TNFs (mean rate: 2.51 (CI 95% 1.7–3.4/100 pts-years; *p* = 0.02). and malignancy rates were higher on VDZ/UST (mean rate: 2.14 (CI 95% 1.6–2.8)/100 pts-years; *p* = 0.01). **Conclusions:** Rates of AEs and infections were not different among biologicals. Infusion/injection reactions were more common on anti-TNFs. Current data are insufficient to suggest the sequencing of biologicals in elderly patients based on safety.

## 1. Introduction

Inflammatory bowel diseases (IBD), ulcerative colitis (UC) and Crohn’s disease (CD) are chronic immune-mediated disorders of the gastrointestinal tract that can negatively impact patients’ physical health and quality of life [1].

Biologic therapies have revolutionized medical management for IBD over the past two decades and are associated with improved outcomes. Biologic therapies are immunomodulatory drugs, and their use may be associated with multiple adverse events (AEs), including infusion/injection reactions, infections and/or malignancies, as reported in landmark clinical trials and post-marketing registries. However, they are still understudied in some specific patient populations, such as the elderly, who may be more vulnerable to AEs [2,3].

Currently, biologic therapy for elderly patients with IBD broadly follows the same algorithms as for younger patients with IBD. Nevertheless, there is no consensus on the suggested sequencing among biologics in the elderly based on safety and/or efficacy, although the available data do not suggest a decrease in clinical efficacy [4,5]. 

IBD therapy for elderly patients may be challenging compared to younger patients due to their advanced age and increased comorbidities, as well as polypharmacy and age-related changes in pharmacokinetics [6]. It is known that treatment with immunosuppressants and/or biologics may confer a risk of infection and malignancy, which could be even more relevant in elderly patients. However, underutilization of these therapies may be associated with poorer outcomes [7,8].

There is still a lack of data on the efficacy and safety profile of anti-TNF agents and newer biologics in the elderly, most published trials did not focus on IBD patients older than 60 years. Indications for biologicals in the elderly are similar to those for younger patients [9]. Data on the real-world safety of biologics in elderly patients are available mainly from rheumatology case series, and many are controversial [10,11,12].

Our aim was to perform a systematic review and meta-analysis to assess the safety of approved biologic therapies in the elderly IBD population in real-world studies reporting rates of AEs, infections, infusion/injection reactions and malignancies.

## 2. Materials and Methods

### 2.1. Data Sources and Search Strategy

A systematic review on the safety of biological therapies in elderly patients with IBD was conducted in PUBMED and SCOPUS between 1 April 1969 and 1 September 2021, as well as conference annals proceedings (World Congress of Gastroenterology, American College of Gastroenterology, Canadian Digestive Disease Week, Digestive Disease Week and United European Gastroenterology Week) between 1 January 2011 and 1 September 2021. In addition to searching study references and reviewing articles, we contacted authors for additional information. To search Medline for safety in elderly IBD patients, we used string #1 (Inflammatory Bowel Disease [MeSH] OR Crohn’s disease [MeSH] OR Ulcerative Colitis [MeSH] OR ‘Crohn’s disease’ [ti] OR ‘Ulcerative Colitis’[ti]), AND #2 (‘elderly’ [ti]), AND #3 (‘patient’ [ti]), AND #4 (‘biologic’ [ti] or ‘infliximab’ [ti] or ‘adalimumab’ [ti] or ‘vedolizumab’ [ti] or ‘ustekinumab’ [ti] or ‘tofacitinib’ [ti] or ‘small molecule’ [ti]), AND #5 (‘side effect’ [ti] or ‘safety [ti] or ‘adverse event’ [ti] or ‘infection’ [ti] or ‘cancer’ [ti] or ‘lymphoma’ [ti]).

The search strategies for safety of treatment of elderly patients with IBD in the biologic era are outlined in Figure 1. An electronic search was conducted by 2 independent reviewers (G.D.H. and P.A.G.) who were blinded to the results of the other search result; any disagreement was resolved by a third reviewer (principal investigator, P.L.L.). Our database search yielded 2885 articles and 12 congress abstracts, with 12 peer-reviewed papers and 3 abstracts meeting our inclusion criteria. Most of the studies were retrospective, integrating IBD patients, with an age limit of 60 or 65 years for elderly, from Europe or North America. The gender ratio was equal, except for the USA veteran database. The identified studies collected safety data on anti-TNF therapy, vedolizumab and ustekinumab. Studies with a follow-up period of at least 1 year were included. 

This study was conducted in accordance with the preferred reporting items for systematic reviews and meta-analyses (PRISMA) statement [13].

### 2.2. Study Selection

We included studies that investigated adult IBD populations (diagnosis of UC or CD per conventional definitions), specifically adult-onset elderly and/or elderly-onset patients, biologic treatment strategies and safety in elderly patients, the rates of adverse events, infections, infusion/injection reactions and cancer rates.

We excluded studies that were not related to our outcome of interest or that were not written in English. We included cohort, case–control, and cross-sectional studies.

### 2.3. Data Extraction

We used a standardized data extraction form. Extracted variables included (I) study characteristics (first author, period of study, location, study design and measurement tools); (II) patient characteristics (age, gender and IBD subtype (UC or CD)); (III) adult-onset elderly and elderly-onset patients, disease duration, biologic treatment duration, line of therapy, previous exposure to biologics and safety in elderly patients, as well as the rates of adverse events, infections, infusion/injection reactions and cancer.

### 2.4. Assessment of Study Quality

The methodological quality of the included studies was assessed using the Newcastle–Ottawa scale (NOS) [14]. Studies were scored across three categories—selection, comparability of study groups and ascertainment of the outcome of interest—with a maximum of two points per questions in certain categories (Appendix A). Study quality was defined as low, moderate or high based on scores of 0–3, 4–6 and 7–9, respectively. Two reviewers (G.D.H. and P.A.G.) independently performed data abstraction and quality assessment. 

### 2.5. Statistical Analysis

Analysis was performed separately for AEs, infections, infusion/injection reactions and malignancy by comparing rates of study outcomes in elderly patients based on the type of biologic used (anti-TNF, VDZ and UST). Heterogeneity between the studies was calculated using Cochrane’s Q and I^2^ statistics. An I^2^ > 50% or *p* < 0.10 indicated significant heterogeneity. 

Publication bias in relation to the incidences of the variables AEs, infection, infusion/injection reaction and malignancy was visually examined using a funnel graph, as well as Begg and Egger tests. To verify the distribution of data and data transformation was required to approximate normality and ensure the power of statistical tests, the Shapiro–Wilk statistical test was used. In the case of non-normality, we opted for least squares transformation, which promotes the minimization of the sum of squares of the model residue.

A significance level of *p* <0.05 was set for rejection of the null hypothesis of normality. Software R Studio version 4.0.4 was used for these analyses.

## 3. Results

### 3.1. Study Characteristics

The characteristics of the eligible studies are summarized in Table 1. In total, the included studies comprised 1978 elderly IBD patients on biologic therapy: 841 on anti-TNF, 816 on VDZ and 321 on UST. Ten of fifteen studies reported on adverse events (AEs) in the elderly, whereas all fifteen studies reported infection rates. Nine studies assessed the rates of infusion/injection reactions, and eleven studies reported cancer rates among elderly IBD patients. The age cutoff was >60 years for 10 studies and >65 years for other 5 studies. Median disease duration ranged between 5 and 20.6 years. Biologic therapy median duration ranged between 1 and 2 years. Concomitant use of steroids was observed in twelve studies, and the rates ranged between 25 and 70% of the included patients. Previous exposure to biologics was reported in eight studies, and the rates varied between 6 and 95% of included patients. Most of the studies were retrospective cohorts (nine studies), three studies were prospective cohorts, two studies were retrospective case–control studies, and one study comprised a retrospective and prospective cohort. Among the eight studies on anti-TNF, seven evaluated elderly patients on IFX and ADA, whereas one evaluated only IFX; six studies evaluated elderly patients on VDZ and three on UST.

### 3.2. Adverse Event (AE) Rates 

Appendix A shows the individual AE rates reported in the articles included in the systematic review. 

As shown in Figure 2, the rates of AE were not statistically significant different according to the type of biologic (mean rate: 11.3 (95% CI 9.9–12.7)/100 PY; *p* = 0.11) in elderly IBD patients treated with anti-TNF, VDZ and/or UST. There was significant heterogeneity between the studies (I^2^ 87%, *p* < 0.01), and the individual rates ranged from 2.42 to 27.19.

The publication bias with respect to the rates of adverse events (AEs) is shown in Appendix A, and no evidence for significant heterogeneity was observed (*t* = −0.20, df = 9, *p*-value = 0.85). 

### 3.3. Infection Rates 

The individual infection rates reported in articles included in the systematic review are summarized in Appendix A.

As shown in Figure 3, infection rates were not statistically significantly different among the studied biologic therapies in elderly IBD patients (mean rate: 9.49 (95% CI 8.4–10.58)/100 PY); *p* = 0.56. There was significant heterogeneity between the studies (I^2^ 90%, *p* < 0.01), and the individual infection rates ranged from 1.83 to 36.90.

Appendix A presents the Funnel plot analysis for detection rates of infection bias. There was no evidence of significant heterogeneity (*t* = −0.07, df = 15, *p*-value = 0.94).

### 3.4. Infusion/Injection Reactions 

The individual infusion/injection reaction rates reported in the articles included in the systematic review are shown in Appendix A.

As shown in Figure 4, the infusion/injection reaction rates were more common in patients on anti-TNFs (*p* < 0.01), although the heterogeneity between the studies was high (I^2^ 83%, *p* < 0.01), and the individual infusion/injection reaction rates ranged from 0.00 to 14.04. 

Appendix A shows a funnel plot analysis with respect to publication bias for infusion/injection reaction. There was no evidence of significant heterogeneity (*t* = −0.32, df = 9, *p*-value = 0.7542).

### 3.5. Malignancy Rates 

Appendix A shows the distribution of individual malignancy rates reported in the articles included in the systematic review. 

As shown in Figure 5, malignancy rates were higher in elderly patients on VDZ and UST (mean rate: 2.14 (CI 95% 1.6–2.8)/100 pts years. The heterogeneity between the studies was moderate (I^2^ 46%, *p* = 0.03), and the individual malignancy rates ranged from 0.67 to 5.26. 

Appendix A presents the results of the funnel plot analysis with respect to publication bias in terms of malignancy rates. As the data do not follow a normal distribution, a transformation to approximate normality was performed to enable statistical analysis (*t* = −1.47, df = 12, *p*-value = 0.16). After data transformation, no significant heterogeneity was observed (*t* = −0.57, df = 18, *p*-value = 0.5789). 

### 3.6. Studies Directly Comparing Anti-TNF and VDZ Efficacy and Safety

The efficacy and safety of anti-TNFs and VDZ were directly compared in two studies (Adar et al. 2019 and Pabla et al. 2021) [17,22]. The earlier study showed no differences in terms of safety profile between the two biologics. In the anti-TNF group, 113 patients (86%) were anti-TNF-naïve, and 123 (94%) were VDZ-naïve. In the VDZ group, 41 patients (40%) were anti-TNF-naïve. Infections were observed in 20% of patients on anti-TNF and 17% on VDZ after 1 year of follow-up (*p* = 0.54) [17].

The latter study included 212 patients (108 on VDZ and 104 on anti-TNF). In the VDZ group, 79 patients (73.2%) had previously failed anti-TNF therapy, along with 18 patients (17.3%) in the Anti-TNF group. No significant differences between the two cohorts in terms of serious infections, surgical intervention or IBD-hospitalization-free survival rates were observed in this study (*p* = NS) [22].

## 4. Discussion

The major finding of our meta-analysis was that the safety of different biologicals, i.e. the rates of AEs and infections, was not different across the investigated biologicals in elderly IBD populations. In contrast, infusion/injection reactions were more common in patients treated with anti-TNFs, and a higher number of malignancies was observed among patients using VDZ and UST. Furthermore, it is important to report that two studies directly comparing anti-TNFs and VDZ reported similar efficacy and safety in the elderly IBD population.

One of the first meta-analyses regarding the efficacy and safety of anti-TNF in IBD patients was that published by Peyrin-Biroulet et al. (2008) [30]. In the overall analysis, the authors reported that there was no difference in terms of frequency of malignancy between anti-TNF-treated patients and the control groups (0.24% vs. 0.39%, respectively; 95% CI, 0.45–0.18). Similarly, there was no difference in the frequency of death between anti-TNF and control groups (0.21% vs. 0.05%, respectively; 95% CI 0.21–0.29) related to malignancies. With regard to infections, there was no difference in the frequency of serious infections between anti-TNF and control groups (2.09% vs. 2.13%, respectively; 95% CI 0.45–0.65) [30]. In another meta-analysis on serious infections in non-elderly IBD patients, Wheat CL et al. showed that no treatment strategy resulted in higher odds of serious infection (including placebo), although in many cases, the confidence intervals were wide, probably partly due to the small study cohort sizes on the different therapies; and the authors concluded that they could not exclude an increase in risk [31]. Lichtenstein et al. reported similar mortality in patients treated with an anti-TNF and those who received other treatments only, although an increased risk of infections was observed in patients treated with IFX [32].

Busquets et al. were among the first to perform a systematic review on the efficacy and safety of anti-TNFs in the elderly, although mainly in patients with rheumatic diseases. The authors concluded, with a low-to-moderate level of evidence, that elderly patients on anti-TNF therapy experience more AEs and similar efficacy when compared with younger patients [5]. Lobatón et al. investigated the efficacy and safety of anti-TNF therapy in elderly and non-elderly IBD patients. Short-term clinical response to anti-TNFs at 10 weeks after anti-TNF initiation was found to be significantly worse in elderly IBD patients (68% vs. 89%; *p* < 0.001), meaning that the probability of ceasing treatment during the follow-up period (regardless of the reason) was higher; however, when primary nonresponse was excluded, this proportion was similar between the two groups. No differences were found in long term efficacy among the initial responders (79.5% vs. 82.8%; *p* = 0.64). With respect to safety, a higher risk of SAEs was observed in elderly IBD patients treated with anti-TNFs (RR = 4.7; *p* < 0.001) compared to the younger subgroup. However, this risk varied according to the type of SAE when elderly patients on anti-TNF were compared to elderly patients on other treatments [16]. A recent study by Calafat et al. compared the rates of IFX-related immune-mediated AEs and loss of response (LOR) in elderly and younger patients. A total of 939 (12%) who started IFX over 60 years of age and 6844 (88%) below 50 years of age were included. Elderly patients presented with a higher proportion of AEs related to IFX (23.2% vs. 19%; *p* = 0.002), infections (7.1% vs. 4.3%; *p* < 0.001) and neoplasms (2.2% vs. 0.5%; *p* < 0.001). In contrast, the rates of immune-mediated AEs (14.8% vs. 14.8%, *p* = 0.999) and infusion reactions (8.1% vs. 8.1%, *p* = 0.989) were alike between the two groups. Based on these results, the authors concluded that elderly patients with IBD have a similar risk of developing IFX-related immune-mediated AEs and LOR compared to younger patients [33]. Additionally, the safety of golimumab and certolizumab in the elderly IBD population was investigated, but no evidence was found. Piovani, D et al. [34] recently conducted a systematic review with meta-analysis and observed an almost threefold higher risk of serious infections and, similarly, opportunistic infections among elderly patients with IBD exposed to biologics than among bio-naïve patients. With regard to the risk of any infection in elderly IBD patients exposed to biologics, no significant increase was observed. In addition, no association was found between cancer and exposure to biologics. In the present systematic review and meta-analysis, we demonstrated that the rates of AEs and infections in anti-TNF-treated patients were not different compared to the other classes biologicals in elderly IBD patients; however, as expected, infusion/injection reactions were more common in patients on anti TNFs. More patients treated with newer biological classes were exposed to an earlier biological therapy; thus, there was variation in terms of the line of biological therapy (see Table 1).

The efficacy and safety of anti-TNFs have been extensively studied, although few real-world or comparative data are available for new biologicals. In landmark clinical trials, they appeared to be a safer option compared to anti-TNFs, although in indirect comparisons. Recently, comparative efficacy and safety data became available in IBD patients. The VARSITY trial compared VDZ vs. ADAL in patients with moderately to severely active, mainly bio-naïve ulcerative colitis patients. There were numerical differences in reported AEs. The exposure-adjusted incidence rate of infection was 23.4 per 100 PY in the VDZ group and 34.6 per 100 PY in the ADAL group [35]. Another recent clinical study, the SEAVUE study, compared UST vs. ADAL for induction and maintenance of biologically naïve patients with moderate-to-severe CD. Regarding safety, 34.0% of UST-treated and 40.5% of ADAL-treated patients had infections, 2.6% and 7.2% had SAEs, and 6.3% and 11.3% had AEs, leading to discontinuation of therapy in non-elderly IBD patients [36]. 

As for the elderly IBD population on new biologicals, there is still a paucity of data concerning efficacy and safety from real-world studies. In 2021, Garg et al. were the first to report on the safety and efficacy of UST in elderly CD patients. The efficacy and safety of UST were similar in this relatively small patient cohort in elderly and non-elderly IBD patients, although elderly patients were less likely to achieve complete clinical and steroid-free remission, and both groups had 95% earlier biological exposure. In addition, the mucosal healing rates observed in the elderly cohort were in line with other real-world studies performed in non-elderly IBD patients [27]. Two studies were included in our meta-analysis regarding efficacy. Adar et al. reported that more anti-TNF-treated CD patients were in remission at 3 months compared to VDZ-treated patients (OR 2.82, 1.18–6.76 CI 95%), although the difference was not maintained at 6 and 12 months [17]. Pabla et al. reported that drug sustainability was superior in patients treated with VDZ (*p* = 0.02) at the end of the 1.4-year study follow-up period (51.9% vs. 45.2%) [22]. As for safety, UST use in elderly IBD was not associated with higher rates of infusion reaction, infections or postoperative complications as compared to non-elderly patients [27]. Interestingly, the present study showed no difference in AEs and infection rates among elderly IBD patients treated with anti-TNF, VDZ and UST, although infusion/injection reaction rates were lower in elderly IBD patients treated with VDZ and UST. In addition, a higher rate of malignancies was observed in patients using VDZ or UST; however, this may represent a selection bias phenomenon, namely that the treating physician may be more likely to start UST or VDZ in patients with a high risk for malignancy based on the beneficial safety profile of the new biologicals reported in landmark clinical trials.

The strength of the present study is that it represents the first systematic review and meta-analysis of real-word studies on comparative safety of biological therapies in the elderly IBD population with a complex and robust analysis of safety outcomes. Additionally, median treatment duration was similar among the studies, which may help to avoid bias due to higher event rates during induction therapy. However, our study is subject to some limitations. First, significant heterogeneity was present among the studies, particularly in terms of the rates of adverse events and infections, which may relate partly to differences in definitions used to describe AEs. Secondly, the reported frequency of each of the AEs might have been biased by the retrospective design (both under- or overreporting are possible) and the interpretation of the events by the treating physician based on the known side effect profile of the specific biological (e.g., higher likelihood of reporting infectious AE for anti-TNFs in the electronic medical record as a related complication). Third, there was a paucity of data on comorbidities in elderly patients, and there was significant variation in terms of concomitant steroid use (without the dose), which could have influenced specific AE rates, e.g. infections. In addition, the definition of SAE differed significantly; some studies did not even highlight SAE and only reported AE rates. Thus, we were not able to formally analyze SAE rates. Finally, variations in disease duration and probability of earlier biological exposure were observed, although this could be more relevant in terms of efficacy rather than safety comparisons among the studies.

## 5. Conclusions

In conclusion, we found that the rates of AEs and infections were not different among the investigated biologicals in elderly IBD patients. Current data are still insufficient to suggest sequencing among biologicals in the elderly based on safety, and larger studies in this specific population are warranted. Comorbidities and potential risk assessment of AEs and infections should be evaluated individually in elderly IBD patients before starting any biological therapy. 

## Figures and Tables

**Figure 1 jcm-11-04422-f001:**
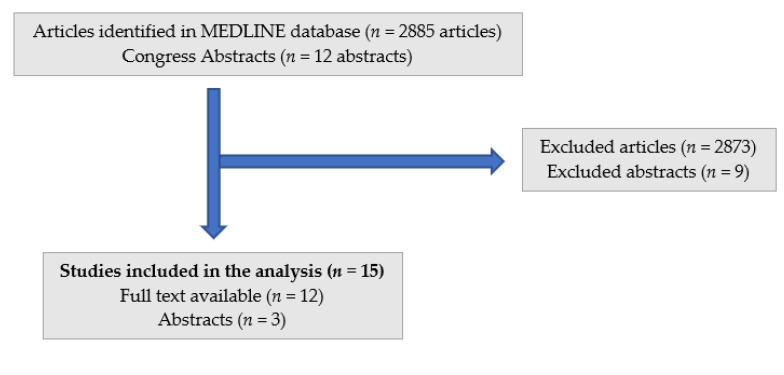
Flow chart of study selection.

**Figure 2 jcm-11-04422-f002:**
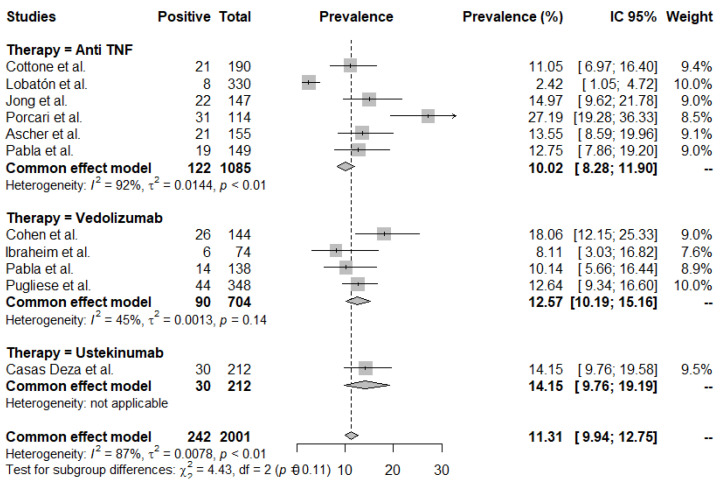
Rates of adverse events (AE). Mean rates per 100 patient years (PY).

**Figure 3 jcm-11-04422-f003:**
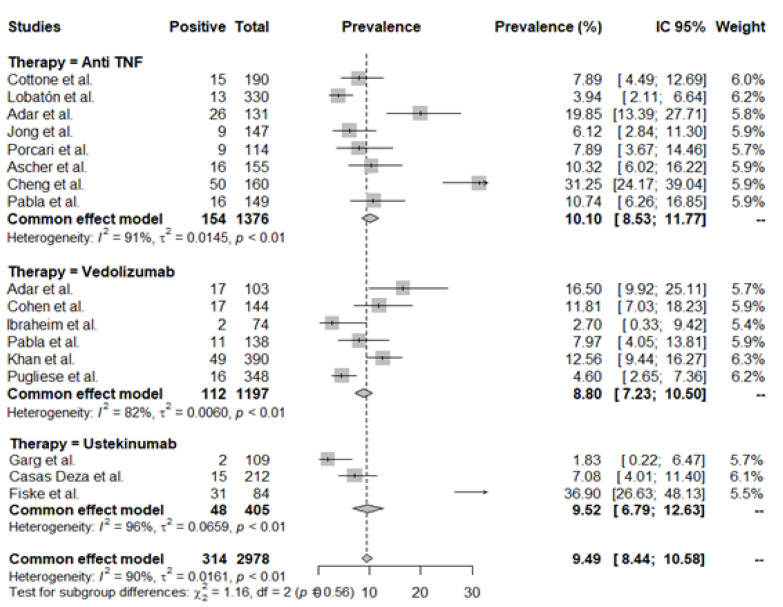
Rates of infection. Mean rates per 100 patient years (PY).

**Figure 4 jcm-11-04422-f004:**
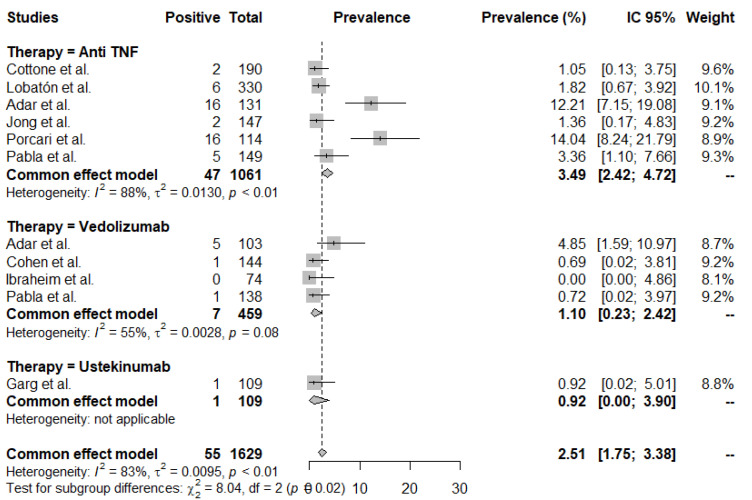
Rates of infusion/injection reaction. Mean rates per 100 patient years (PY).

**Figure 5 jcm-11-04422-f005:**
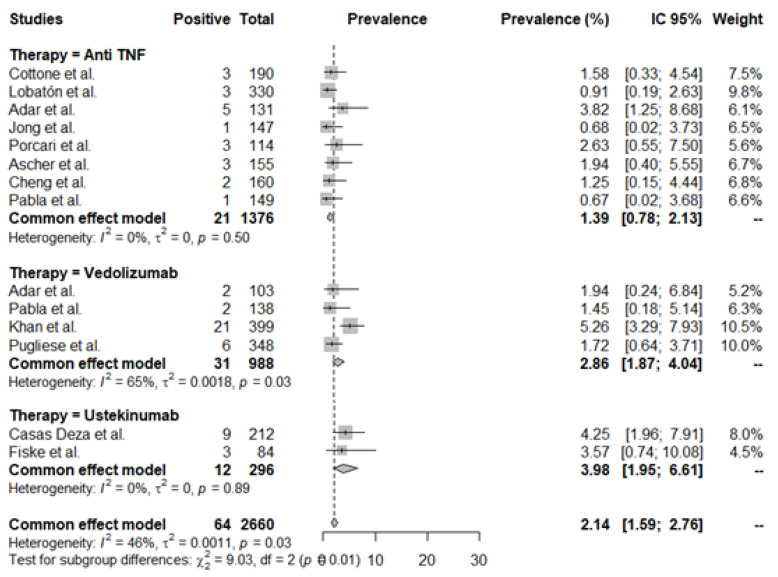
Malignancy rates. Mean rates per 100 patient years (PY).

**Table 1 jcm-11-04422-t001:** Characteristics of studies included in the systematic review and meta-analysis.

Type of Biologic	Year	Author	Age Cutoff	Study Design	Number of Elderly Patients	Median Disease Duration (Years)	Median Treatment Duration (Years)	Previous Biologic Exposure (%)
**Anti-TNF**	2011	Cottone et al. [15]	>65 y	Cohort—Prospective	95	NA	1.4	NA
	2015	Lobatón et al. [16]	>65 y	Case Control—Retrospective	66	6	1	NA
	2019	Adar et al. [17]	>60 y	Case Control—Retrospective	131	13	1	14 (Anti-TNF) 6 (VDZ)
	2020	Jong et al. [18]	>60 y	Cohort—Prospective	81	7.5	1.7	NA
	2020	Porcari et al. [19]	>60 y	Cohort—Retrospective	114	>5	1	NA
	2020	Asscher et al. [20]	>60 y	Cohort—Retrospective	90	16.7	1.7	NA
	2021	Cheng et al. [21]	>60 y	Cohort—Retrospective	160	5.5	1	NA
	2021	Pabla et al. [22]	>60 y	Cohort—Retrospective	104	10	1.43	17.3 (Anti-TNF)
**Vedolizumab**	2019	Adar et al. [17]	>60 y	Case Control—Retrospective	103	16	1	60 (Anti-TNF)
	2020	Cohen et al. [23]	>60 y	Cohort—Retrospective	144	10	1	46.
	2020	Ibraheim et al. [24]	>60 y	Cohort—Retrospective	74	9	1	27
	2021	Pabla et al. [22]	>60 y	Cohort—Retrospective	108	15.5	1.27	73.2 (Anti-TNF)
	2021	Khan et al. [25]	>65 y	Cohort—Retrospective	213	NA	1	NA
	2021	Pugliese et al. [26]	>65 y	Cohort—Prospective	174	10.9	2	55
**Ustekinumab**	2021	Garg et al. [27]	>65 y	Cohort—Retrospective	39	20.6	1.3	95
	2021	Casas Deza et al. [28]	>60 y	Cohort—Prospective/ Retrospective	212	NA	1	85
	2021	Fiske et al. [29]	>60 y	Cohort—Retrospective	70	NA	1	84.3

## Data Availability

The main data are presented in this article. The data are available from the corresponding author upon request.

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
