# Peer review of "Safety of Biological Therapies in Elderly Inflammatory Bowel Diseases: A Systematic Review and Meta-Analysis"

_jcm, 2022, doi:10.3390/jcm11154422_

Round 1

Reviewer 1 Report

This is a systematic review and meta-analysis on the safety of biologics in elderly patients with IBD based on real-world studies. Almost two thousand elderly (aged above 60-65 years) patients were included. Rates of overall adverse events (AEs), infusion/injections reactions, and malignancies were analyzed during anti-TNF, vedolizumab (VDZ), and ustekinumab (UST) treatment. Biologic therapy's median duration ranged between 1 and 2 years. Significant heterogeneity was found between the studies. No different rates of overall AEs and infections were found among the biologics. Infusion/injection reaction rates were more common on anti-TNFs, and malignancy rates were higher on VDZ or UST.

IBD elderly patients are increasing worldwide and they are systematically excluded from RCT, efficacy and safety data are urgently warranted. Hahn’s work aims to fill the safety knowledge gap. The authors discuss adequately the background, methods, results, strengths and limitations of their work. They compare their results with different studies design (elderly vs young IBD patients) and conclude with limited but practice advice.

I suggest the authors to:

a.     Analyze and compare AE rates in IBD subtype (CD and UC) - at least as supplementary

b.     Expand treatment duration and its cumulative effect on AE rates.

c.     Further discuss the most important study results difference (e.g. Lobaton and Porcari in AE, Adar, Cheng, and Fiske in infections, Adar and Porcari in infusion reactions) - e.g comparing definitions and study design

d.     Report types and numbers of SAE for each biologic – even rare, but severe, events can influence the safety profile

e.     Please correct the authors' affiliations (number 6 doesn’t exist)

Author Response

Response to reviewers

We thank the reviewers for their work and for their important comments and questions that helped us to further improve the paper.

Please find below our detailed responses to the reviewer #1 comments:

Reviewer #1

This is a systematic review and meta-analysis on the safety of biologics in elderly patients with IBD based on real-world studies. Almost two thousand elderly (aged above 60-65 years) patients were included. Rates of overall adverse events (AEs), infusion/injections reactions, and malignancies were analyzed during anti-TNF, vedolizumab (VDZ), and ustekinumab (UST) treatment. Biologic therapy's median duration ranged between 1 and 2 years. Significant heterogeneity was found between the studies. No different rates of overall AEs and infections were found among the biologics. Infusion/injection reaction rates were more common on anti-TNFs, and malignancy rates were higher on VDZ or UST.

IBD elderly patients are increasing worldwide and they are systematically excluded from RCT, efficacy and safety data are urgently warranted. Hahn’s work aims to fill the safety knowledge gap. The authors discuss adequately the background, methods, results, strengths and limitations of their work. They compare their results with different studies design (elderly vs young IBD patients) and conclude with limited but practice advice.

I suggest the authors to:

a. Analyze and compare AE rates in IBD subtype (CD and UC) - at least as supplementary

Thank you for the comment. We reported the articles in their totality, unfortunately, such data could not be retrieved in all of them, so it was decided not to compare by subtype due to lack of data for that.

b. Expand treatment duration and its cumulative effect on AE rates.

Thank you for the comment. The articles founded and included evaluated the effects in at least 12 months of treatment, although to expand treatment duration would be interesting to evaluate cumulative effects on AE rates for future research.

c. Further discuss the most important study results difference (e.g. Lobaton and Porcari in AE, Adar, Cheng, and Fiske in infections, Adar and Porcari in infusion reactions) - e.g comparing definitions and study design

Thank you for the comment. We reported the articles in their totality, further evaluation on different outcomes could not be retrieved in all of them due to lack of data reported. Heterogeneity was present among the studies, particularly for the rates of adverse events and infections, which may relate partly to differences in definitions used to describe AEs.

d. Report types and numbers of SAE for each biologic – even rare, but severe, events can influence the safety profile

Thank you for the comment, unfortunately definition of SAE differed significantly, some studies did not even highlight SAE, just reported AE rates, thus we were not able to analyze formally SAE rates. Heterogeneity was present among the studies, particularly for the rates of AE’s and infections, which may relate partly to differences in the definitions used.

e. Please correct the authors' affiliations (number 6 doesn’t exist)

Thank you for the comment, it was corrected (track changes)

We hope that our responses are adequate to further clarify and improve our study and we thank the reviewers for their comments.

Reviewer 2 Report

In this systematic review and meta-analysis, the authors assessed the safety of biological therapies in elderly IBD. The manuscript is well written and the authors did a good job presenting the results in a relevant manner.

Comment 1: I think the medline search might have been sub-optimal. I don’t think variable #3 “patients” was necessary and including it with an “AND” might have filtered out some papers. Variable #4 Biologics etc. could have included more terms such as “golimumab” “certolizumab pegol” “anti-integrin” “tumor necrosis factor” “anti-adhesion”. It’s true that the ones included in the paper are the “higher-yield” entries but for completion, I think these should have been included.

Comment 2: Typo under materials and methods. AND “#2” patient should be “#3”.

Comment 3: 2.5 Statistical Analysis “malignancy and were” is confusing – change the word “were” to “was” or rephrase the sentence.

Comment 4: I suggest that this relatively recent study by Piovani D et al be included in the discussion as it tackles a very similar topic and can add relevance to your manuscript. Piovani D, et al. Systematic review with meta-analysis: biologics and risk of infection or cancer in elderly patients with inflammatory bowel disease. Aliment Pharmacol Ther. 2020;51(9):820-830.

Author Response

Response to reviewers

We thank the reviewers for their work and for their important comments and questions that helped us to further improve the paper.

Please find below our detailed responses to the reviewer #2 comments:

Reviewer #2

In this systematic review and meta-analysis, the authors assessed the safety of biological therapies in elderly IBD. The manuscript is well written and the authors did a good job presenting the results in a relevant manner.

Comment 1: I think the medline search might have been sub-optimal. I don’t think variable #3 “patients” was necessary and including it with an “AND” might have filtered out some papers. Variable #4 Biologics etc. could have included more terms such as “golimumab” “certolizumab pegol” “anti-integrin” “tumor necrosis factor” “anti-adhesion”. It’s true that the ones included in the paper are the “higher-yield” entries but for completion, I think these should have been included.

We appreciate the comment and your thorough evaluation on that. It will be taken into consideration for further research. Additionally, safety of Golimumab and Certolizumab in the elderly IBD population were also searched, but no evidence was found.

Comment 2: Typo under materials and methods. AND “#2” patient should be “#3”.

Thank you for the comment. It was corrected (track changes).

Comment 3: 2.5 Statistical Analysis “malignancy and were” is confusing – change the word “were” to “was” or rephrase the sentence.

Thank you for the comment. The sentence was rephrased as suggested (track changes).

Comment 4: I suggest that this relatively recent study by Piovani D et al be included in the discussion as it tackles a very similar topic and can add relevance to your manuscript. Piovani D, et al. Systematic review with meta-analysis: biologics and risk of infection or cancer in elderly patients with inflammatory bowel disease. Aliment Pharmacol Ther. 2020;51(9):820-830.

We appreciated the comment and suggestion. The mentioned article was added in our discussion section (track changes).

We hope that our responses are adequate to further clarify and improve our study and we thank the reviewers for their comments.